# Cross-Cultural Adaptation and Validation of the Functional, Communicative and Critical Health Literacy Instrument (FCCHL-SR) for Diabetic Patients in Serbia

**DOI:** 10.3390/healthcare10091667

**Published:** 2022-08-31

**Authors:** Marija Levic, Natasa Bogavac-Stanojevic, Dusanka Krajnovic

**Affiliations:** 1Department of Social Pharmacy and Pharmaceutical Legislation, Faculty of Pharmacy, University of Belgrade, Belgrade 11221, Serbia; 2Department of Medical Biochemistry, Faculty of Pharmacy, University of Belgrade, Belgrade 11221, Serbia

**Keywords:** translation and cultural adaptation, confirmatory factor analysis, perception-based outcome measurement instrument, generic scale, self-reported, subjective measurement, chronic non-infectious diseases

## Abstract

Thoroughly validated instruments can provide a more accurate and reliable picture of how the instrument works and of the level of health literacy in people with type 2 diabetes mellitus (T2DM). The present work aimed at cross-cultural adaptation and validation of the Functional, Communicative and Critical Health Literacy Instrument (FCCHL) in patients with T2DM in Serbia. After translation and back-translation, views from an expert group, one cognitive interview study (*n* = 10) and one survey study (*n* = 130) were conducted among samples of diabetic patients. Item analysis, internal consistency, content validity, confirmatory factor analysis (CFA) and reliability testing were performed. When all 14 items were analyzed, loading factors were above 0.55, but without adequate model fit. After removing two items with the lowest loadings FHL1 and IHL2 the fit indexes indicated a reasonable normed χ^2^ (SB scaled χ^2^/df = 1.90). CFI was 0.916 with SRMR = 0.0676 and RMSEA = 0.0831. To determine internal consistency, Cronbach’s alpha coefficient was 0.796 for the whole FCCHL-SR12. With only minor modifications compared to the English version, the 12-item FCCHL instrument is valid and reliable and can be used to measure health literacy among Serbian diabetic patients. However, future research on a larger population in Serbia is necessary for measuring the levels of HL and their relationship with other determinants in this country.

## 1. Introduction

During the last three decades, the importance of Health Literacy (HL) and optimal health outcomes has been recognized [1,2,3,4,5]. HL has been given a prominent place in some important documents issued by the World Health Organization (WHO) and the European Union (EU) [6,7]. There are several definitions and conceptual models of HL [8,9,10,11], the most commonly cited definition is from 2000 where Ratzan and all define HL as: “the degree to which individuals have the capacity to obtain, process, and understand basic health information and the services needed to make appropriate health decisions” [12]. 

The definition of HL was revised in August 2020 with the publication of the U.S. Government’s Healthy People 2030 external icon initiative. Audit involves the division of HL into personal HL and organizational HL and provides the following definitions: Personal health literacy is the degree to which individuals have the ability to find, understand and use information and services to inform health decisions and actions for themselves and others. Organizational health literacy is the degree to which organizations fairly enable individuals to find, understand, and use information and services to inform health decisions and actions for themselves and others [13].

Although there are different definitions of health literacy it has been proven that people with low levels of health literacy have less compliance with medical information and drugs, increased but inefficient use of the health system, more visits to the emergency center, higher use of drugs, and a higher risk of death [14,15,16,17]. In addition to the negative effects, low health literacy is both an economic burden on society and an alarming public health problem [2]. Among the elderly, people with low socio-economic status and minority groups, the greater presence of low health literacy has been reported, which significantly contributes to health inequalities [16]. Low health literacy is linked directly or indirectly to a large number of poor health outcomes. Data show that there is a correlation between low health literacy and reduced use of available health information and services. This is reflected in a greater need for health education and use of preventive health services [18,19]. 

It is extremely important to properly measure HL skills in order to gain insights into the level of patients’ HL. However, so far, a lot of available instruments show several problems. First, they usually have to be used by a healthcare professional, which is time consuming and impracticable in clinical practice. Second, the basic constructions and the content of existing instruments varies, and only a few instruments are based on the proposed definitions and models of health literacy. Finally, most existing HL measures are focused primarily on understanding reading, while health literacy considers more than functional literacy, namely abilities for constructive use of information [20,21].

A theoretical model which is cited in the professional literature and useful in analyzing the literacy abilities required in various health situations is the Nutbeam model. This model distinguishes three types of health literacy: functional (FHL), communicative/interactive (IHL) and critical health literacy (CHL). Each of these types of health literacy requires different skills for obtaining, understanding, and using information. FHL represents the basic level of reading and writing necessary for living effectively in everyday situations. IHL considers more advanced cognitive and writing skills, which, together with social skills, allow people to extract information, derive meaning from various forms of communication and apply new information when circumstances change. CHL presents more advanced skills for analysis of data from critical perspective and using information to exert greater control over life events and situations [22]. 

Ishikawa et al. developed a HL self-assessment instrument (Functional, Communicative and Critical Health Literacy scale–FCCHL) which relies on this model and has the aim to measure all three types of HL. It has been recognized as one of the most suitable and comprehensive instruments for measuring health literacy in people with diabetes in healthcare settings [23,24]. Patients with diabetes and limited health literacy often cannot read medication labels accurately, may take medication incorrectly, have less medication adherence, and generally have difficulty understanding instructions for follow-up care [25,26]. These patients also have poorer patient-doctor communications and participate less in decision-making [27].

Altin et al. found out that most HL scales could be deemed multidimensional. The use of multidimensional scales in health-related research far outweighs the number of published studies that apply multidimensional analyses approaches. Multidimensional scale like FCCHL uses subscales to measure different but related aspects in order to capture the complexity of a construct. Multidimensional modeling approaches are appropriate to account for the observed covariance in the data [28,29].

FCCHL has been validated in several populations including French/Dutch/German/Australian/Japanese/Norwegian citizens [21,30,31,32,33,34,35]. However, no validation of FCCHL exists in Serbian. Validated translations of HL measures are needed, as a growing literature has shown the importance of evaluating HL in patients with type 2 diabetes mellitus (T2DM) [21,36,37,38,39,40]. Permission to use the FCCHL was obtained from the author (Hirono Ishikawa) under e-mail agreement (9 January 2020) and we used the English version of the FCCHL, which includes 14 items.

There is limited knowledge of Functional, Communicative, Critical and total HL in Serbia, and so far, there has been no validated instrument for measuring all these health literacy levels. Due to a nature of the disease and large distribution of the DMT2 population in Serbia it is of exceptional importance to identify patients’ needs and work on improvement of disease control and quality of life of this population. Thus, the aim of this article is to describe the process of translation, cultural adaptation, and validation of the FCCHL instrument into Serbian in order to make it suitable to be used in Serbian healthcare settings. 

## 2. Materials and Methods

### 2.1. Instrument

The FCCHL is a general perception-based instrument, that is a subjective measure involving respondents to rate their perceived abilities. Across three levels (F-functional, I-communicative (interactive), and C-critical) with answer categories ranging from 1 (never) to 4 (frequent). This self-reported instrument consists of 14 items. FHL1-FHL5, measures reading comprehension. IHL1-IHL5, assess skills in finding, understanding, and applying information and communicating personal views on diabetes. Four items, CHL1-CHL4, critically assess the ability to self-report by assessing the reliability, validity, and applicability of available health-related information. Scores on the functional HL scale were recorded, and mean scores were calculated for each scale ranging from 1 (low health literacy) to 4 (high health literacy) [35]. The current FCCHL does not define cut-off or class values for health literacy. 

### 2.2. Translation and Cultural Adaptation

At the beginning of the preparation for the research, before the validation procedure, it is necessary to adjust the instrument to the language in which the research is conducted, as well as to the population of the participants.

Experts of the International Society for Pharmacoeconomics and Outcomes Research (ISPOR) have set guidelines that define the basic principles of translation and adaptation of the instrument: (1) translation preparation, (2) “forward” translation, (3) single “forward” translation, (4) “backward” translation, (5) review of the “backwards” translations, (6) harmonization, (7) cognitive examination, (8) review and (9) final report [41,42]. 

In preparation for translation, people were selected to do the translation (A1, A2, T3 and T4). The methodology is defined, and the author of the instrument was contacted to gain approval for use of the FCCHL instrument.“Forward” translation in our case was the translation of the instrument from the source language (English) into target language (Serbian). This step was performed by two -researches (A1 and A2) whose native language is Serbian, and the other language is the source language of the scale being translated. Both authors were familiar with the concept of the research. They were independent of each other, i.e., all items, answers and instructions were translated separately. When translating, focus was maintained on ensuring that the concept is adequately conveyed and that the wording is clear.Single “forward” translation or the formation of a unified version of the translation involved merging these two researches into one (A12) and this was done by a third person from the team and after discussion between the researchers. This version was with a minimum of disagreement and with the clearest questions in translations.“Backward” translation was done by translating from target language into the source language. It was conducted by two translators (T3 and T4) who are native speakers of the source language and are fluent in the target language. Both back translators were unfamiliar with the content of the instrument.A review of the “backwards” translations considered a comparison of back-translated versions of an instrument with the original to highlight and explore the differences between the original and the aligned translation.The harmonization implies a central place in the whole process and involved comparison of both versions of the “backwards” translations, testing the degree of agreement of the concepts of all items, making corrections, controlling language errors, and forming a version for the testing phase.The penultimate step in the cultural adaptation process is pre-testing. It is a process in which the final version was introduced into testing on the population for which the instrument was made. Pre-testing was done using the cognitive interviewing technique “probing” with required patients at a health-care institution by a researcher (A1) [29,43]. To gain a better understanding of the cognitive processes the participant used to answer the items thinking aloud, as explicitly instructed. Ten diabetic patients were eligible to fill-in the instrument and discuss it with the interviewer. Interviews were conducted until data saturation was reached; meaning that no more new information of value was obtained. It lasted from 5–6 min.In the review process all reports from previous stages were reviewed in detail, the test results were included in the translation and all disagreements were eliminated. The degree of equality between the target version and the original was assessed, and the result of this step is the creation of the final version of the instrument.The final report considered a review of the final version of the instrument and submission of reports with all collected documents to the author. The authors evaluated and approved the final version of the FCCHL to be used for the validation study. (Figure 1).

### 2.3. Quantitative Study

The quantitative study (validation study) was used to evaluate the reliability, structural validity, distributional properties, and convergent validity of the FCCHL-SR14 instrument.

### 2.4. Sample and Data Collection

The target population of the validation study were patients diagnosed with T2DM at least six months before the start of the study, who knew the Serbian language, aged 18 and older and voluntarily agreed to participate with signed informed consent. The exclusion criteria were participants with medical background (e.g., doctors, study nurses, pharmacists…) and those who provided less than 90% of answers in the instrument. In total, we approached 147 persons, out of which approximately 88% fulfilled the study criteria. We excluded 17 individuals due to not fulfilling 90% of the instrument. The final sample for validation study included 130 individuals. The sample size is often dependent on the length of the instrument, as some authors recommend that the participant-to-item ratio should be at a minimum 5:1 [44]. Larger sample sizes could provide more meaningful factor loadings and factors and yield more generalizable results, so we opted for a participant-to-item ratio of 10:1.

This study was carried out at one healthcare center and one community pharmacy randomly chosen from two different municipalities in the Belgrade region. Patients from all parts of those municipalities were represented to reflect the geographical distribution in the target population. Data for this cross-sectional study were collected between January 2021 and June 2021 and between March and April 2022, using a self-administered paper-and-pencil instrument. Before the survey, we recruited five research assistants to help us with collecting data. To ensure that they were familiar with the purpose, process, and procedure of applying the instrument, we systematically trained three pharmacy graduates and two doctors as research assistants. Throughout data collection, the researchers and assistants explained the purpose and significance of the study to the participants and obtained written informed consent. Participants did not receive any payment for filling out the instrument. All data was anonymous and, as such, entered into the database. 

Demographic variables were collected, such as gender, age, education level, self-reported general health condition, life habits and questions related to diabetes.

### 2.5. Data Analysis

We used mean value and standard deviation (SD) for normally distributed data, median and 25. and 75. percentile values for skewed data and absolute and relative frequencies to characterize the study sample. Also, we calculated FCCHL total scores and domain scores. Normality of distribution was tested by the Kolmogorov Smirnov test. To describe the FCCHL we also analyzed minimal and maximal values for each item. Distributional properties of the instrument (skewness and kurtosis) were further inspected to examine the normality of the scores on each subscale and to identify floor and ceiling effects. Floor or ceiling effects were considered to be present if >15% of the patients scored the worst or the best possible score [45]. 

The comparative fit index (CFI), the standardized root mean square residual (SRMR), and the root mean square error of approximation (RMSEA) were used to examine the model fit. Normed χ^2^ < 3, CFI values ≥ 0.95 and SRMR and RMSEA values < 0.08 and ≤ 0.06, respectively, were considered indicative of good model fit [46]. However, RMSEA values of 0.08 could indicate an acceptable fit [47,48]. Factor loadings over 0.71 were considered excellent, 0.63 very good, and 0.55 good [45]. To improve the model with inadequate fit, e.g., when CFI, SRMR, and RMSEA were unsatisfactory, we examined the modification index (MI) and allowed to correlate measurement errors, or we removed items with the lowest factor loadings. We compared the first and final model by computing a χ^2^ difference test to assess incremental fit. According to this test and recalculated coefficients, we decided whether a new models fit significantly better than the given model. 

After confirming the instrument’s validity, reliability was assessed by internal consistency and test-retest methods. In the internal consistency method, consistency of the results of the tool items was investigated, and then the Cronbach’s alpha coefficient was calculated for the items in each domain and the whole instrument. Test-retest reliability or consistency in answering items was examined by asking 29 patients with T2DM who participated in the validation process to refill the same instrument after four weeks. Interclass correlation coefficient (ICC) was calculated for the items in each domain and the whole instrument. Overall, *p* values less than 0.05 were considered significant [49,50].

All analyses were performed using IBM SPSS Statistics for Windows, Version 27.0. Armonk, NY, USA: IBM Corp except for CFA. It was conducted by Jamovi Statistical Software (Idaho State University).

### 2.6. Ethical Considerations

This study was approved by the Ethics committee of the Healthcare Centre “Zvezdara” (ref. no. 4411-3) and by the Ethics committee of the Pharmacy “Filly farm”. Participation was voluntary, and the instrument was completed anonymously.

## 3. Results

### 3.1. Report of Translation

During the translation process, minor issues were identified by the third person who was involved in the review of “forward” translations and a consensus version was agreed between authors and the reviewer of the translation before re-translation to the source language. During the reconciliation process, the researches have accepted the use of Serbian translation for “diabetes (sugar disease)” covering the word “diabetes” in English, with the aim to explain the medical term to the participants. 

### 3.2. Pre-Testing

The mean age of interviewed participants in the first pre-test was 62.7 years (SD = 12.4), ranging from 34 to 79 years of age. Of the 10 respondents, just over half were men (60%). 50% had completed education at a higher school, university, or university PhD level, and about the same proportion had completed primary and secondary school. Participants primarily lived in urban areas (60%) and on average it took them 3 min to complete the instrument.

No item was considered irrelevant by the participants. Examples of the input of the respondents’ comments during the development of the FCCHL-SR14 instrument are introduced in Table 1 (cultural adaptation) and Table 2 (linguistic adaptation).

The form of the instrument was adjusted based on the advice of a few participants, who did not manage at first that there were 14 separate items for the three categories with four answers offered (Never, Rarely, Sometimes and Often), the font was increased, and it was decided to be in the form of landscape so that elderly can also read with ease.

### 3.3. Subjects

In Table 3, sample characteristics of the validation study are shown. Mean age was 58.2 years with 63.8% of the sample being female. On average, patients have had T2DM for 11 years.

### 3.4. Distributional Properties 

Items in IHL and CHL domains showed no skewness or kurtosis in the distribution of scores. One item in FHL domain (small print) kurtosis was negative and indicated the small outliers in a distribution (Table 4). There was no floor (14.6% FHL; 12.3% CHL; 10.8% IHL, respectively) or ceiling effects in each HL score (4.6% FHL; 8.5%CHL; 9.2%IHL, respectively).

### 3.5. Structural Validity and Reliability and Suggested Modifications to the FCCHL-SR14

Structural validity was examined by CFA. When we analyzed all 14 items (FCCHL-SR14), loading factors were between 0.49 and 0.77 (Table 4), but without adequate model fit (Table 5). Examining MI of the unique-error terms, we found that two correlated-error terms had MIs greater than 10-one for FHL questions (between 1st item (FHL1) “Found that the print is too small to read even though you wear glasses” and 2nd item (FHL2) “Found unfamiliar characters and words” (MI was 19.2) and one for IHL questions (between the 1st item (IHL1) “Extracted only information you wanted” and the 2nd item (IHL2) “Collected information from different sources” (MI was 14.8). We rerun the FCCHL-SR14 model, first freeing the largest correlated error and, after that, the second. As seen in Table 5, the modified FCCHL-SR14 model fit the data significantly better when it included the one correlated-error terms with the largest MI (MI or ∆χ^2^ = 19.2; *p* < 0.001). Although the model’s fit coefficients were improved, its CFI was still below 0.90. We thus freed the other correlated error with MI = 14.8 and reestimated the model. The model including two correlated-error terms significantly improved the model’s fit (∆χ^2^ = 14.8; *p* < 0.001) but still without appropriate fit coefficients. In the next step, we examined the factor loadings for each of the two pairs of questions that shared measurement error to remove the question with the lower factor loading. Questions with shared measurement error in FHL domain were FHL1 and FHL2 with factor loadings of 0.439 and 0.643, respectively; in IHL domain were IHL1 and IHL2 with factor loadings of 0.556 and 0.431, respectively. 

After removing two items with the lowest loadings, FHL1 (Found that the print is too small to read even though you wear glasses) and IHL2 (Extracted (only) information you wanted) in the modified FCCHL-SR12 the fit indexes indicated a reasonable normed χ^2^ (SB scaled χ^2^/df = 1.88). As seen in Table 5, FCCHL-SR12 was not worse than FCCHL-SR14 (χ^2^ difference *p* value was < 0.001). CFI, SRMR, and RMSEA for FCCHL-SR12 indicated a good model fit and the second model was retained. Standardized factor loadings ranged between 0.54 and 0.79 for the correlated 3-factor model of the HL scales in the total sample (*n* = 130). Rectangles represent the observed variables (items) and ellipses represent the hypothesized latent constructs (factors). Values on the single-headed arrows leading from the factors to the items are standardized factor loadings. Values to the left of the items represent error variances. Values on the curved double-headed arrows are correlations between factor terms. (Figure 2).

The highest subscale correlation was observed between the IHL and CHL subscales (r = 0.851). Independent of the modeling approach, the lowest factor loadings were observed for the items FHL1 and IHL2.

The FCCHL-SR12 instrument was assessed by internal consistency and test-retest methods. To determine internal consistency, Cronbach’s alpha coefficient in a sample of 130 patients was 0.767 for the whole FCCHL-SR12 with 95% confidence intervals from 0.703 to 0.822. However, this value varied from 0.792, 0.748, and 0.796 for functional, communicative, and critical constructs, respectively.

To determine the instrument’s consistency in the repeatability dimension, in a group of 29 patients with four weeks’ interval, the ICC for the whole instrument was calculated to be 0.981 with 95% confidence intervals (0.960–0.991). This value varied from 0.980 to 0.960 and 0.972 in functional, communicative, and critical domains, respectively

## 4. Discussion

### 4.1. Cultural and Linguistic Adaptation of the FCCHL-SR Instrument

Like in other studies investigating the FCCHL [29,30,31,32,33,34,35] our results indicate that, after translating and adapting the FCCHL instrument to Serbian, the FCCHL-SR12 is a valid instrument, ready to be used in Serbia, and opening possibilities to study HL in Serbia and compare the results internationally.

We found that inclusion of lay people helped a lot in designing and simplification of the instrument, for being consistent with the broad and inclusive definition of HL. The pre-testing was an important step in the translation process, which eventually led to the Serbian version of FCCHL. Even though the specialist review turned out to be essential regarding accepted language within the health and social setting, the pre-testing gave vital information about the understanding of actual people who might answer the instrument. Including the target audience when translating instruments to another language and their influence on the adaptation is crucial for creation of a valid and reliable instrument to be used in clinical practice settings.

Patients with T2DM perceived some difficulties in filling out the items. Some items left room for interpretation, and additional clarification/examples were provided to give patients a better idea of the concept. 

### 4.2. The 12-Item FCCHL-SR

Similar to the study in Norway [34], FCCHL-SR12 has several benefits over the FCCHL-SR14 version. The FCCHL-SR12 has a better normed χ^2^, CFI, SRMR and RMSA, and the remaining FHL and IHL items had a better fit to the model. 

Respondents who stated in FHL1 that they “found that the print was too small to read” could indicate their opinions about the font size, font type, or their sight variables-which might be independent of HL. After pre-testing, the item was rephrased with the addition “even though you wear glasses” and in this way, the item was better clarified. In addition, in IHL2-“extracted information you wanted” confused participants since it is too general, and they suggested adding “only” in between. Considering the lowest factor loadings of these two items and unclarities among participants, they were removed after discussion. 

### 4.3. Methodological Considerations 

In accordance with previous studies [29,30,33], exploratory analysis revealed a 3-factor model confirming the overall structure of the scale, with satisfactory internal consistency of each FCCHL dimension. 

Regarding the distributional properties of the instrument, there were no floor or ceiling effects in each HL score, the same as in some other studies [31,33,51], which shows that we cannot expect a distribution problem with lower ability to differentiate people with very low and very high health literacy levels. We have the same results for distribution of scores with previous findings on this instrument [29].

The instrument showed a good internal consistency (Cronbach’s alpha coefficient = 0.84, 0.77 and 0.65 respectively) in the study from Ishikawa [35], and its three-level structure looked promising for the measurement of the full spectrum of HL. Our findings differ slightly from previous findings in the Netherlands [29] and Australia [32], which found that internal consistency of the communicative dimension was less satisfactory (α  =  0.63 in both studies). However, due to this difference, the instrument should be further investigated in larger samples. 

The subscale correlation was observed between the IHL and CHL subscales, which suggests that the measurement of IHL can be substituted for the measurement of CHL. As FHL is defined as basic skills, while communicative HL and critical HL are defined as advanced skills [29], use of FCCHL-SR12 instrument may contribute to promoting a better understanding of advanced skills beyond reading comprehension and numeracy.

Responding to self-administered measures could be quite challenging for people with limited FHL since it requires reading and reading comprehension abilities. However, the participants reported that the items were clearly stated, while they were being interviewed.

This study provides evidence for the reliability and validity of the FCCHL-SR12. 

### 4.4. Advantages of FCCHL Scale

While other scales focus on functional health literacy, this scale aims to measure the broader concept of health literacy, including the ability to retrieve, understand, and use health-related information. 

Health literacy has been presented as a measurable and important concept in considering education for patients with chronic diseases such as diabetes. In addition to the previous instruments that focus exclusively on functional health literacy, this scale covers all three levels of health literacy, each of which can have different effects on patient outcomes. Also, the scale is easy to apply in clinical conditions.

Exploring the functional, communicative, and critical levels of patients’ health literacy can help physicians and other health care workers to better understand their patients’ potential barriers to disease self-management and health-promoting behaviors [36].

### 4.5. Limitations

HL was assessed with a self-report instrument which could lead to social desirability and an overestimation of the HL level, as individuals are often ashamed of their inability to read. The study can be performed in a larger population.

## 5. Conclusions

The FCCHL scale was selected for translation, adaptation, and validation because it is short, easy to administer, and it is the only instrument for health literacy which measures individually functional, communicative, and critical health literacy as well as the total health literacy. The findings indicated that the Serbian version of FCCHL (FCCHL-SR12) is comparable to the original model and according to the model fit, a three-dimensional approach (where the correlations between the subscales are taken into account) is recommended when using the FCCHL to describe HL in people with T2DM. This opens possibilities to study HL at health-care settings in Serbia and internationally compare the results. The specialist review and pre-testing provided essential additional information to the translation/back-translation procedure. Adaptations that were made helped to bring the instrument closer to the target group. FCCHL-SR12 demonstrated adequate reliability and validity as an internal measure for Serbian patients with T2DM. This validated model might be helpful in the countries where there is a lack of validated tools for measuring HL levels. Future research on a larger population in Serbia is necessary in order to draw conclusions about the levels of HL and their relationship with other determinants in this country.

## Figures and Tables

**Figure 1 healthcare-10-01667-f001:**
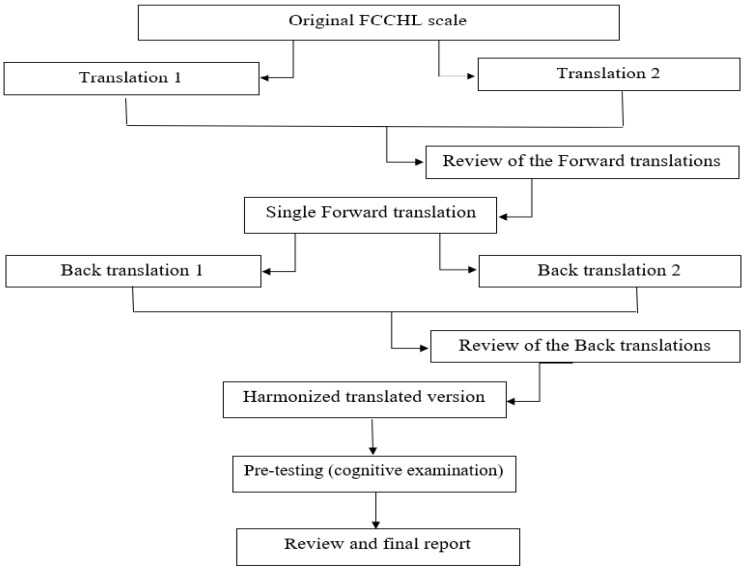
Translation and cross-cultural adaptation steps for FCCHL instrument.

**Figure 2 healthcare-10-01667-f002:**
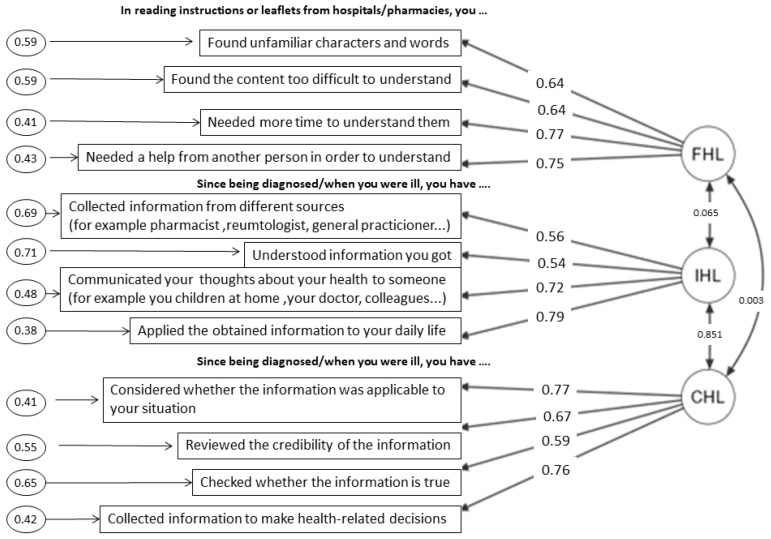
Summary of Structural Validity.

**Table 1 healthcare-10-01667-t001:** Cultural adaptation of the items from the FCCHL-SR14 instrument included in the discussion after pre-testing.

	Initial Variant of the Item	Suggestions after Pre-Testing	Changes
FHL1	Found that the print is too small to read	It was unclear for respondents whether it is applicable in the case of wearing glasses Suggestion: to add ‘even with glasses’	Found that the print is too small to read even though you wear glasses
FHL4	Found the content too difficult	It was unclear what it meant to be too difficult Suggestion: to add ‘to understand’	Found the content too difficult to understand
FHL5	Needed someone to help you read them		Needed help from another person in order to understand
IHL1	Collected information from different sources	Respondents were not sure what the different sources represent Suggestion: to add examples	Collected information from different sources (for example pharmacist, rheumatologist, general practitioner...)
IHL2	Extracted the information you wanted	It was unclear for participants what this item presents Suggestion: to clarify with adding ‘only’	Extracted (only) information you wanted
IHL4	Communicated your thoughts about your health to someone	Respondents were confused by the term someone Suggestion: To clarify the term with examples	Communicated your thoughts about your health to someone (for example you children at home, your doctor, colleagues...)

**Table 2 healthcare-10-01667-t002:** Linguistic adaptation of the items from the FCCHL-SR14 instrument included in the discussion after pre-testing.

	Initial Variant of the Item	Suggestions after Pre-Testing	Changes
FHL2	Found characters and words that you did not know	Rephrased from “did not know” to “unfamiliar” for better understanding	Found unfamiliar characters and words
CHL4	Collected information to make decisions about your health	Changed to be in the spirit of the language	Collected information to make health-related decisions

**Table 3 healthcare-10-01667-t003:** Characteristic of 130 participants in the validation study.

	*n* (%)
**Marital status**	
Unmarried	15 (11.5%)
Married/Common-law	85 (65.4%)
Divorced	17 (13.1%)
Widow	13 (10%)
**Children**	
Yes	102 (78.5)
No	28 (21.5)
**Number of children**	
One child	30 (24.6)
Two children	57 (46.7)
Three or more children	14 (11.4)
**Education**	
4 classes or no school	1 (0.8%)
Primary school	5 (3.8%)
High school	44 (33.8%)
Higher school (VI grade)	29 (22.3%)
University	48 (36.9%)
Master’s degree/Specialization/PhD grade	3 (2.3%)
**Employment**	
Incapable	2 (1.5 %)
Unemployed	10 (7.7 %)
Student	1 (0.8 %)
Employed	77 (59.2 %)
Pensioner	39 (30.0 %)
**Monthly income per family member**	
≤27,000 RSD *	16 (12.3%)
27,000–40,000 RSD	22 (16.9%)
≥40,000–60,000 RSD	86 (66.2%)
≥60,000 RSD	6 (4.6%)
**Chronic diseases**	
T2DM T2DM and additional chronic diseases	43 (33%) 87 (67%)
**Therapy for T2DM**	
Diet	1 (0.8 %)
Tablets	83 (63.8 %)
Tablets and Insulin	36 (27.7 %)
Insulin	10 (7.7 %)
**Frequency of drug administration for T2DM**	
Once a day	8 (6.2%)
Twice a day	69 (53.1%)
Three times a day	37 (28.5%)
Four times a day	15 (11.5%)
I don’t use drugs for T2DM	1 (0.8%)
**Active exercise**	
Never	27 (20.8%)
Less than once a week	46 (35.4%)
1–2 times a week	37 (28.5%)
3 and more times a week	20 (15.4%)
**Smoker**	
≤1 box a day	35 (26.9%)
>1 box a day	18 (13.8%)
Not smoker	68 (52.3%)
Ex-smoker	9 (6.9%)
**Alcohol**	
Never	74 (56.9%)
Once a month	35 (26.9%)
2 or more times a month	21 (16.2%)
**Source of health information**	
Doctors	67 (51.5 %)
Pharmacists	9 (6.9 %)
Parents	1 (0.8 %)
Internet	18 (13.8 %)
Friends	1 (0.8 %)
Books/Magazines/TV	3 (2.3 %)
Doctors and Pharmacists	27 (20.8 %)
Doctors and Internet	1 (0.8 %)
Doctors, Pharmacists, and Internet	3 (2.3 %)
**Interest in health**	
Not interested	3 (2.3%)
Little	22 (16.9%)
Medium	66 (50.8%)
Much	21 (16.2%)
Very interested	18 (13.8%)
**Self-estimation of health status**	
Very bad	6 (4.6 %)
Bad	31 (23.8 %)
Good	77 (59.2 %)
Very good	16 (12.3 %)

Note. * 1 RSD = 0.0085 EUR.

**Table 4 healthcare-10-01667-t004:** Distribution of scores.

FHL	(1) Small Print	(2) Unfamiliar Characters and Words	(3) Difficult Content	(4) More Time Needed	(5) Needed Help
Mean	2.05	2.17	2.32	2.19	2.51
Median	2.00	2.00	2.00	2.00	3.00
Standard deviation	0.951	0.916	0.856	0.872	0.950
Skewness	0.331	0.147	0.077	0.183	−0.022
Kurtosis	−1.05	−1.00	−0.653	−0.756	−0.899
Standardized factor loadings	0.543	0.722	0.641	0.733	0.689
IHL	(1) Information sources	(2) Wanted information	(3) Understanding the information gathered	(4) Sharing thoughts with someone	(5) Application of information
Mean	2.48	2.52	2.78	2.79	2.60
Median	2.00	2.50	3.00	3.00	3.00
Standard deviation	0.865	0.799	0.853	0.938	0.886
Skewness	0.048	0.062	−0.177	−0.202	−0.003
Kurtosis	−0.628	−0.436	−0.660	−0.929	−0.734
Standardized factor loadings	0.599	0.490	0.549	0.696	0.756
CHL	(1) Considered the applicability of the information	(2) Credibility of information	(3) Checking the accuracy of information	(4) Collecting information	
Mean	2.72	2.47	2.48	2.65	
Median	3.00	2.00	3.00	3.00	
Standard deviation	0.872	0.873	0.837	0.929	
Skewness	−0.283	−0.011	−0.071	−0.019	
Kurtosis	−0.543	−0.663	−0.550	−0.911	
Standardized factor loadings	0.772	0.675	0.604	0.752	

**Table 5 healthcare-10-01667-t005:** Models fit coefficients.

Model	χ^2^	df	*p*	CFI	SRMR	RMSEA (90%CI)
FCCHL-SR14	192	74	<0.001	0.819	0.0779	0.111 0.092–0.130
Modified FCCHL-SR14 with one correlated error	173	73	<0.001	0.846	0.0753	0.103 0.084–0.123
Modified FCCHL-SR14 with two correlated error	158	72	<0.001	0.867	0.0731	0.0961 0.0761–0.117
FCCHL-SR12	96	51	<0.001	0.916	0.0676	0.0831 (0.057–0.108)
Δ FCCHL-SR14-FCCHL-SR12	96	23	<0.001			

## Data Availability

The datasets generated during and/or analyzed during the current study are available from the corresponding author upon reasonable request.

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
