# Peer review of "Cross-Cultural Adaptation and Validation of the Functional, Communicative and Critical Health Literacy Instrument (FCCHL-SR) for Diabetic Patients in Serbia"

_healthcare, 2022, doi:10.3390/healthcare10091667_

Round 1

Reviewer 1 Report

Healthcare-1796759

Comments to the author

Title: Communicative and Critical Health Literacy Instrument (FCCHL-SR) For Diabetic Patients in Serbia

-       The study was aimed at cross-cultural adaptation and validation of the FCCHL in patients with T2DM in Serbia. Generally, the methodology used in this study seems not sufficiently elaborated. In addition, some explanations or words are lengthy and redundant, so they need to be more concise and clearer.

-       The acronyms such as "FCCHL" should be used after full words.  

-       The importance of HL from a health perspective could be explained better systematically. Please revise it to be more concise and structured.

-       “This is especially important for people with inadequate literacy skills who have a lower level of understanding of written and oral information to adhere to medical regimens and scheduled regular.”: This statement needs proper evidence because people with lower HL tend to show low medication compliance.  

-       Please state the importance of HL for DM patients and briefly introduce the trend of previous studies related to the issue. In addition to the need for a Servian version instrument, the contribution of this study to the bulk of previous knowledge needs to be clarified as well.  

-       “Validated translations of HL measures are needed, as a growing literature has shown the importance of evaluating HL in patients with type 2 diabetes mellitus (T2DM).”: Authors need to explain the reason or importance with supportive references.

-       Translation and back-translation procedures seem to be properly implemented. However, too lengthy. Please revise to be concise. Also please make clear in using the terms source language and target languages in both different steps to avoid confusing.

-       Please make clear the statement that “exclusion criteria were medical background.” What kind of medical background?

-       A response providing less than 90% of answers can be excluded from the analysis. However, incomplete response is not a kind of exclusion criteria of “participants.” Please make clear about it.

-       Is there any reason of choosing one healthcare center and one pharmacy? The sampling method seems a convenient sampling method. Please state it clearly.

-       The method for model comparison is lacking.

-       P11. Authors removed two items, FHL1 and IHL2 with lowest factor loadings from the model. However, authors did not provide significant test results comparing two models. Having low factor loading is not a sufficient condition to remove items.

-       Fig 2:

n  On the left side, error variances are provided in rectangles. However, an error term needs to be expressed as a round shape because it is a latent term. In addition, the figures in the rectangles seem incorrect because the sum of the squared factor loading (standardized) and error variance of an item should be one. Please make them reasonable.

n  The correlation between FHL and IHL, and FHL and CHL, are extremely small, compared with other studies. Authors need to explain the specific reason clearly.

-       Table 12. instrument was assessed”: “Table 12.” seems a typological error. Please correct.

-       Cronbach's alpha coefficient in a sample of 130 patients was 0.796 for the whole FCCHL-SR12 with 95% confidence intervals: Can not find the actual 95% CI.

-       Authors need to justify the removal of two items FHL1 and IHL2, because different studies have removed different items each other.

-       “Since the 12-item version contains different numbers of items with a predominance of items reflecting CHL…”: What does this statement mean? Because this 12-item version contained 4 items in each dimension, there is no predominance of any one dimension.

-       There seems that serious problems in generalization of the results because the sample carries only 130 T2DM Servian patients with mean age of 58.2 years. The limitation in generalization is clear because patients with various diseases, community population, or people with different cultural background.      

Author Response

Dear Reviewer 1, 

Thank you so much for your valuable comments, please find here our responses and corrected work attached.

-       The acronyms such as "FCCHL" should be used after full words.  

Thank you for your comment, the abbreviation FCCHL is explained in the Abstract, under the line number 15.

-       The importance of HL from a health perspective could be explained better systematically. Please revise it to be more concise and structured.

Thank you for suggestion, the importance of HL has been revised in the third paragraph of the Introduction.

-       “This is especially important for people with inadequate literacy skills who have a lower level of understanding of written and oral information to adhere to medical regimens and scheduled regular.”: This statement needs proper evidence because people with lower HL tend to show low medication compliance.  

Thank you for your suggestion, the whole paragraph of the Introduction has been revised, also covered under the previous observation and explanation.

-       Please state the importance of HL for DM patients and briefly introduce the trend of previous studies related to the issue. In addition to the need for a Serbian version instrument, the contribution of this study to the bulk of previous knowledge needs to be clarified as well.  

Thank you for your valuable comment, importance of HL for DM patients with trends are presented under the lines 100-104. There is a limited knowledge on a level of HL in Serbia for DM2 patients and due to a nature of the disease and large distribution of the DMT2 population in Serbia it is of exceptional importance to identify patients' needs and work on improvement of disease control and quality of life of this population, added to the lines 120-122.

-       “Validated translations of HL measures are needed, as a growing literature has shown the importance of evaluating HL in patients with type 2 diabetes mellitus (T2DM).”: Authors need to explain the reason or importance with supportive references.

Thank you for your suggestion, the refences are added under the line 115.

-       Translation and back-translation procedures seem to be properly implemented. However, too lengthy. Please revise to be concise. Also please make clear in using the terms source language and target languages in both different steps to avoid confusing.

Thank you for your suggestion, the process of translation and cultural adaptation has been revised accordingly and the terms source language and target languages clearly indicated, to avoid possible confusion.

-       Please make clear the statement that “exclusion criteria were medical background.” What kind of medical background?

Thank you for the comment, it does consider doctors, study nurses, pharmacists… It is added under the lines 217-218.

-       A response providing less than 90% of answers can be excluded from the analysis. However, incomplete response is not a kind of exclusion criteria of “participants.” Please make clear about it.

Thank you, it is clarified under the lines 218-219.

-       Is there any reason of choosing one healthcare center and one pharmacy? The sampling method seems a convenient sampling method. Please state it clearly.

-       The method for model comparison is lacking.

Thank you for your valuable comment, it is added, lines 256-260.

-       P11. Authors removed two items, FHL1 and IHL2 with lowest factor loadings from the model. However, authors did not provide significant test results comparing two models. Having low factor loading is not a sufficient condition to remove items.

Thank you for the suggestion. We compared two models by chi squared test. In the Data analysis section, we added an explanation of models’ comparison and in the Result section, the new Table 5 with model fit coefficients has been added. We used all additionally calculated data to compare models. 

-      On the left side, error variances are provided in rectangles. However, an error term needs to be expressed as a round shape because it is a ltent term. In addition, the figures in the rectangles seem incorrect because the sum of the squared factor loading (standardized) and error variance of an item should be one. Please make them reasonable.

We apologize for the mistake in the Figure. We corrected the error values and changed the Figure according to the reviewer's suggestions

- The correlation between FHL and IHL, and FHL and CHL, are extremely small, compared with other studies. Authors need to explain the specific reason clearly.

Thank you for your comment, we got these coefficients. This combination of questions best describes the FHL and IHL domains in our country. The fact that the questions differ in different populations indicates that this instrument should be adapted to the population in this way through this validation procedure.

-       “Table 12. instrument was assessed”: “Table 12.” seems a typological error. Please correct.

Thank you for your observation, it is corrected.

-       Cronbach's alpha coefficient in a sample of 130 patients was 0.796 for the whole FCCHL-SR12 with 95% confidence intervals: Cannot find the actual 95% CI.

Thank you for your observation, we added 95%CI interval in the Result section.

-       Authors need to justify the removal of two items FHL1 and IHL2, because different studies have removed different items each other.

Thank you for your valuable comment, in our population, this combination of questions best describes the FHL and IHL domains. Scores in the domain related to FHL and IHL would not have been significantly impaired even if the instrument without the removed questions was used, because that instrument is not worse than the instrument without those two questions, but the validation coefficients are better in the abbreviated version. The fact that the questions differ in different populations indicates that this instrument should be adapted to the population in this way through this validation procedure.

-       “Since the 12-item version contains different numbers of items with a predominance of items reflecting CHL…”: What does this statement mean? Because this 12-item version contained 4 items in each dimension, there is no predominance of any one dimension.

Thank you for your observation, it is corrected.

-       There seems that serious problems in generalization of the results because the sample carries only 130 T2DM Servian patients with mean age of 58.2 years. The limitation in generalization is clear because patients with various diseases, community population, or people with different cultural background.    

The instrument was validated for chronic patients, and we used it for DMT2 patients due to a nature of their disease and wide distribution in our country. We haven’t found in literature that different cultural background has an impact on HL, regardless of this information in Serbia 83% of population is Serbian nationality and only 17% are national minorities (Etnička i starosna struktura stanovništva većih gradova Srbije (slobodnaevropa.org), therefore the sample was representative to be used for research. The patients are randomly chosen in these two institutions. In Serbia children up to 14 years encompass 14.3%, 65.0% of citizens are aged 15–65 years, while seniors over 65 years make 20.7% of the population. The study sample was representative of the population in our country as there were no significant differences in socio-demographic data of questionnaire respondents and the general population of the Republic of Serbia (Republic Statistical Office. Statistical calendar of the Republic of Serbia. 2021 February 5. Available from https://publikacije.stat.gov.rs/G2021/Pdf/G202117014.pdf).

We are looking for to hearing more from you.

All the best,

Authors 

Reviewer 2 Report

Synopsis

Health literacy is an important determinant of health in chronic diabetes patients. Fourteen question items of the Functional, Communicative, and Critical Health Literacy Scale (FCCHL) developed by Ishikawa et al in 2008 in English were translated into Serbian by guidelines set by the Experts of the International Society for Pharmacoeconomics and Outcomes Research. Confirmatory factor analysis was used to validate the Serbian version of FCCHL, in which two items were removed for a better model fit (FCCHL-SR12). Comparative fit index (0.916), Standardized Root Mean Square Residual (0.0676), and Root Mean Square Error of Approximation (0.0831) were reported. The final model structure was represented in Figure 2. Cronbach’s alpha of 0.796 for FCCHL-SR12 indicates a reasonable internal consistency. The study concluded that the FCCHL-SR12 is a valid and reliable instrument to measure health literacy in Serbian diabetic patients.

Reviewer's conflict of interest: None

General comment to authors

Thank you very much for letting me review this well-composed manuscript.  This draft manuscript does not have the line numbers and thus, when pointing out revision suggestions, referencing may be difficult. Figures and Tables do not have a clear ‘title.’ A short description after the label of a figure or table can be a title; however, when the description is longer, it is a caption (footnote). Consider separating the title and caption for each figure or table. For instance, Figure 2 may be titled, ‘Summary of Structural Validity.’

Comments are listed below:

1.      Introduction: The third paragraph describes the importance of health literacy well and the fourth paragraph describes the gap in the scientific development of a measurement tool well.

2.      Section 2.2: Translation and cultural adaption steps are well described with a good reference.

3.      Figure 1 shows the instrument adaptation processes well.

4.      The first paragraph of section 2.4: “The sample size is often dependent on the length of the questionnaire, as some…” should be “…dependent on the number of items in the survey…”

5.      Section 2.5: Data analysis. Were the 29 patients sampled by convenience or randomly  selected from the 130 recruited individuals? Was there any drop-out between the pre- and post-tests?

6.      Section 2.6 should be indented.

7.      How were the 29 test-retest patients tracked if the questionnaire was completely anonymous? Please explain.

8.      Table 3 contains the comprehensive data for demographics. The unit of reporting is not consistent (m, SD) (n, %), or (median, range). Is there a way to make the units consistent?

9.      Please indicate ‘n’ on Table 4. This may be redundant; however, if you can add another row indicating the standardized factor loading, readers can understand why two questions were removed due to loading factors lower than 0.55.

10.  Consider creating a title for Figure 2. The caption for Figure 2 is well composed.

11.  Section 3.5: The 3rd paragraph (just below the Figure 2 caption) has a typo “Table 12.” It should be “Twelve-items in Table 2” or FFCHL-SR12.

12.  Where do the 29 patients belong within the 10 interviewed and 130 surveyed patients? See #5 above.

13.  Discussion is well-organized with topics.

End of review.

Author Response

Dear Reviewer 2, 

Thank you so much for your valuable comments, please find here our responses and corrected work attached.

  1. Introduction: The third paragraph describes the importance of health literacy well and the fourth paragraph describes the gap in the scientific development of a measurement tool well.

Thank you for your observation.

  1. Section 2.2: Translation and cultural adaption steps are well described with a good reference.

Thank you for your comment.

  1. Figure 1 shows the instrument adaptation processes well.

Thank you for your comment.

  1. The first paragraph of section 2.4: “The sample size is often dependent on the length of the questionnaire, as some…” should be “…dependent on the number of items in the survey…”

Thank you for your observation, it is added under the line 223.

  1. Section 2.5: Data analysis. Were the 29 patients sampled by convenience or randomly  selected from the 130 recruited individuals? Was there any drop-out between the pre- and post-tests?

We tried to include a longitudinal follow-up of the study sample of 130 individuals. Only 29 patients accepted to fill in the instrument after one month when come for their therapy.  

  1. Section 2.6 should be indented.

Thank you for your observation, it is corrected.

  1. How were the 29 test-retest patients tracked if the questionnaire was completely anonymous? Please explain.

We asked each participant if he/she would like to do the instrument again in a month when he/she comes for his/her therapy, those who agreed were given special codes and came with them after a month, in this way anonymity was achieved.

  1. Table 3 contains the comprehensive data for demographics. The unit of reporting is not consistent (m, SD) (n, %), or (median, range). Is there a way to make the units consistent?

Thank you for your comment, we removed demographics from the table and made explanation in the text, lines 306 and 307.

  1. Please indicate ‘n’ on Table 4. This may be redundant; however, if you can add another row indicating the standardized factor loading, readers can understand why two questions were removed due to loading factors lower than 0.55.

Thank you for your observation, standardized factor loadings are added to the Table 4.

  1. Consider creating a title for Figure 2. The caption for Figure 2 is well composed.

Thank you for your comment, we implemented your suggestion for the tittle, a description is moved to the main text, lines 358-364.

  1. Section 3.5: The 3rdparagraph (just below the Figure 2 caption) has a typo “Table 12.” It should be “Twelve-items in Table 2” or FFCHL-SR12.

Thank you for your observation, it is corrected to FCCHL-SR12 instrument.

  1. Where do the 29 patients belong within the 10 interviewed and 130 surveyed patients? See #5 above.

29 patents are among 130 individuals included in the sample for validation study. The validation study was quantitative study used to evaluate the reliability, structural validity, distributional properties, and convergent validity of the FCCHL-SR14 instrument.  10 patients that filled-in the instrument and discuss it with the interviewer were included in a pre-validation study - cultural adaptation process, so they participated in pre-testing.  

  1. Discussion is well-organized with topics.

Thank you for your comment.

We are looking forward to hearing more from you.

All the best,

Authors

Round 2

Reviewer 1 Report

Healthcare-1796759-R1

Comments to the author

Title: Communicative and Critical Health Literacy Instrument (FCCHL-SR) For Diabetic Patients in Serbia

Most questions were resolved by the authors’ responses. However, there are more to be clarified or revised.

- Regarding the generalization of results, the authors replied that the proportion of the Serbian population was similar to the composition of the participants. However, a similar proportion does not always mean ‘representative.’ Moreover, the sample lacks people with various diseases.

- The standardized factor loadings for FHL1 and IHL2 were not too small, as 0.543 and 0.490, respectively. The figures are above 0.4, which is the general standard for considering removal. It is questionable that the authors did not use modification indices in the determination of removal.

- Table 5 shows the comparison statistics between the two models, however, the contents lack the statistical test results. Also, the difference in degrees of freedom was 23, only after removing two items.

Author Response

Dear Reviewer 1,

Thank you for your valuable comments, please find below our answers and updated manuscript in the attachment.

Most questions were resolved by the authors’ responses. However, there are more to be clarified or revised.

- Regarding the generalization of results, the authors replied that the proportion of the Serbian population was similar to the composition of the participants. However, a similar proportion does not always mean ‘representative.’ Moreover, the sample lacks people with various diseases.

Thank you for your comment, in the research 33% of participants had only T2DM, the rest of the participants stated that they have at least one more chronic disease, therefore this is not a limitation of this research. This data has been added to the Table 3, characteristic of the patients.

- The standardized factor loadings for FHL1 and IHL2 were not too small, as 0.543 and 0.490, respectively. The figures are above 0.4, which is the general standard for considering removal. It is questionable that the authors did not use modification indices in the determination of removal.

Thank you for the suggestion, this improves our manuscript. In the Result section, we explained how we modified the basic model with 14 items and included information about modification indices. Also, in the Methodology section, we added explanations related to modification indices.

- Table 5 shows the comparison statistics between the two models, however, the contents lack the statistical test results. Also, the difference in degrees of freedom was 23, only after removing two items.

In Table 5, we included statistical test results. The difference in degrees of freedom was 23, and we can confirmed it by manual calculation. DF1 for FCCHL – SR14 was (14x14+1)/2=105 for individual variances and covariances minus 31 (14 individual variances, 14 error variances, 3-factor variances); DF=105-31=74. DF2 for FCCHL – SR12 was (12x(12+1)/2))-27=51. The difference between DF1 and DF2 was 23.

Best regards,

Authors